

1 2 3
# Molecular composition and volatility of isoprene photochemical oxidation secondary organic aerosol under low and high NO$_x$ conditions

*Emma L. D'Ambro[1], Ben H. Lee[2], Jiumeng Liu[3], John E. Shilling[3,4], Cassandra J. Gaston[5],*
*Felipe D. Lopez-Hilfiker[6], Siegfried Schobesberger[2], Rahul A. Zaveri[3], Claudia Mohr[7], Anna*
*Lutz[8], Zhenfa Zhang[9], Avram Gold[9], Jason D. Surratt[9], Jean C. Rivera-Rios[10], Frank N.*
*Keutsch[10], Joel A. Thornton[2]*
[1]Department of Chemistry, University of Washington, Seattle, WA, 98195, USA
[2]Department of Atmospheric Sciences, University of Washington, Seattle, WA, 98195, USA
[3]Atmospheric Sciences and Global Change Division, Pacific Northwest National Laboratory,
Richland, WA, 99352, USA
[4]Environmental Molecular Sciences Laboratory, Pacific Northwest National Laboratory,
Richland, WA, 99352, USA
[5]Rosenstiel School of Marine & Atmospheric Science, University of Miami, FL, 33149, USA
[6]Laboratory of Atmospheric Chemistry, Paul Scherrer Institute, Zurich, Switzerland
[7]Institute of Meteorology and Climate Research, Karlsruhe Institute of Technology, Karlsruhe,
Germany
[8]Department of Chemistry, Atmospheric Science, University of Gothenburg, Gothenburg,
Sweden
[9]Department of Environmental Sciences and Engineering, Gillings School of Global and Public
Health, University of North Carolina, Chapel Hill, NC, 27599, USA
[10]John A. Paulson School of Engineering and Applied Sciences and Department of Chemistry
and Chemical Biology, Harvard University, Cambridge, MA, USA
*Correspondence to*: Joel A. Thornton (thornton@atmos.uw.edu)



**Abstract.** We present measurements of secondary organic aerosol (SOA) formation from
isoprene photochemical oxidation formed in an environmental simulation chamber using dry
neutral seed particles, thereby suppressing the role of acid catalyzed multiphase chemistry, at a
variety of oxidant conditions. A high-resolution time-of-flight chemical ionization mass
spectrometer (HRToF-CIMS) utilizing iodide-adduct ionization coupled to the Filter Inlet for
Gases and AEROsols (FIGAERO) allowed for the simultaneous online sampling of the gas and
particle composition. Under high $HO_2$ and low NO conditions, highly oxygenated (O:C ≥ 1) $C_5$
compounds were major components (~50%) of the SOA. The overall composition of the SOA
evolved both as a function of time and as a function of input NO concentrations. As the level of
input NO increased, organic nitrates increased in both the gas- and particle-phases, but the
dominant non-nitrate particle-phase components monotonically decreased. We use comparisons
of measured and predicted gas-particle partitioning of individual components to assess the
validity of literature-based group-contribution methods for estimating saturation vapor
concentrations. While there is evidence for equilibrium partitioning being achieved on the
chamber residence time scale (5.2 hours) for some individual components, significant errors in
group-contribution methods are revealed. In addition, >30% of the SOA mass, detected as low-
molecular weight compounds, cannot be reconciled with equilibrium partitioning. These
compounds desorb from the FIGAERO at unexpectedly high temperatures given their molecular
composition, indicative of thermal decomposition of effectively lower volatility components,
likely larger molecular weight oligomers. We use these insights from the laboratory and
observations of the same SOA components made during the Southern Oxidant and Aerosol
Study (SOAS) to assess the importance of isoprene photooxidation as a local SOA source.





## 1    Introduction


Atmospheric aerosol particles reduce visibility, adversely affect human health, and have
uncertain overall effects on global climate [*Poschl*, 2005], with particles smaller than 1 μm in
diameter playing important roles. Submicron particles typically contain a significant fraction of
organic material, on the order of 20-90% [*Jimenez et al.*, 2009; *Zhang et al.*, 2007]. Particulate
organic material can be emitted directly to the atmosphere, known as primary organic aerosol, or
formed from the gas-to-particle conversion of volatile organic compound (VOC) oxidation
products which can partition [*Donahue et al.*, 2011; *Riipinen et al.*, 2011] or react
heterogeneously [*Docherty et al.*, 2005; *Gaston et al.*, 2014; *Jang et al.*, 2002; *Surratt et al.*,
2007; *Surratt et al.*, 2006] on existing particles, or homogeneously nucleate to form new
particles [*Kirkby et al.*, 2016]. This condensed phase organic material arising from gas to particle
conversion is known as secondary organic aerosol (SOA).

Biogenic VOC (BVOC) contribute significantly to SOA. Emitted at rates of 500

TgC/year [*Guenther et al.*, 2012] and with a high reactivity, isoprene ($C_5H_8$) has the potential to
contribute substantially to SOA, even if the overall conversion is inefficient. Initially, the
observed products of isoprene oxidation were of high volatility, which led to the hypothesis that
isoprene did not generate SOA [*Pandis et al.*, 1991]. However, subsequent chamber experiments
showed that the yield of SOA from isoprene photochemical oxidation can range from <1-29%
with the highest yields achieved either with acidic aqueous seed particles [*Surratt et al.*, 2010] or
as a transient during successive oxidative aging [*Kroll et al.*, 2006]. Chemically speciated
measurements of atmospheric aerosol components in an isoprene-rich environment identified
polyol compounds likely formed from isoprene oxidation [*Claeys et al.*, 2004; *Paulot et al.*,
2009b]. Subsequent chamber studies have shown that, under low NO conditions, isoprene reacts



with OH followed by HO$_2$ to form a hydroxy hydroperoxide, ISOPOOH, which further reacts
with OH to form the isoprene epoxy diol, IEPOX [*Paulot et al.*, 2009a; *Paulot et al.*, 2009b].
Both laboratory and field studies suggest that IEPOX plays an important role in the formation of
isoprene SOA (iSOA) *via* acid catalyzed heterogeneous reactions on deliquesced particles
[*Gaston et al.*, 2014; *Lin et al.*, 2013a; *Lin et al.*, 2014; *Lin et al.*, 2012; *Lin et al.*, 2013b; *Liu et*
*al.*, 2014; *Nguyen et al.*, 2014; *Paulot et al.*, 2009b; *Surratt et al.*, 2010; *Surratt et al.*, 2006]. In
the absence of acidic seed particles, iSOA yields have generally been low, but functional group
analyses suggested a significant contribution of peroxide moieties and a complex dependence
upon NO$_x$ [*Dommen et al.*, 2006; *King et al.*, 2010; *Kroll et al.*, 2005; 2006; *Sato et al.*, 2011; *Xu*
*et al.*, 2014; *Zhang et al.*, 2011]. Despite these advances, a comprehensive molecular
characterization of photochemical iSOA has been lacking.
Much attention has been focused on the formation of SOA derived from IEPOX
chemistry; however, understanding the formation of SOA from pathways other than IEPOX is
important for quantifying SOA in environments where the SOA is likely formed via other
mechanisms due to the lack of acidic seed. Three recent studies have performed photochemical
oxidation on either ISOPOOH [*Krechmer et al.*, 2015; *Riva et al.*, 2016] or both isoprene and
ISOPOOH [*Liu*, 2016] in the absence of wet acidic seed in order to study the mechanism of
iSOA formation when the IEPOX pathway is suppressed. These studies identified several C$_5$H$_{8-}$
$_{12}$O$_{4-8}$ compounds in both the gas- [*Krechmer et al.*, 2015] and particle- [*Liu*, 2016; *Riva et al.*,
2016] phases. Liu et al. [2016] found that under the photochemical conditions of their chamber,
the most abundant compound in the particle-phase was C$_5$H$_{12}$O$_6$, ISOP(OOH)$_2$, presumed to be a
dihydroxy dihydroperoxide formed from the reaction of an organic peroxy radical (RO$_2$) derived
from ISOPOOH + OH followed by further reaction with hydroperoxyl radicals (HO$_2$) [*Liu*,



2016]. However, the iSOA yields starting from isoprene reported by Liu et al. [2016] were
substantially higher than those starting from ISOPOOH alone as reported by Krechmer et al.
[2015], and generally higher than most previous iSOA studies in the absence of deliquesced
acidic seed particles [*Dommen et al.*, 2006; *King et al.*, 2010; *Xu et al.*, 2014].
Furthermore, there is significant interest in understanding how anthropogenic pollutants
affect SOA yields [*Shilling et al.*, 2013; *Weber et al.*, 2007; *Xu et al.*, 2015], and there have been
several chamber studies to understand the role of $NO_x$ specifically on iSOA yields [*Dommen et*
*al.*, 2006; *King et al.*, 2010; *Kroll et al.*, 2005; 2006; *Xu et al.*, 2014; *Zhang et al.*, 2011]. The
general effect of $NO_x$ on the newly discovered non-IEPOX SOA system has been described
previously [*Liu*, 2016]. The total SOA mass concentration was shown to be stable for input NO
concentrations from 0-20 ppb, with a sharp decrease in SOA mass concentration at the highest
input NO concentration (50 ppb). While these studies have advanced our knowledge of the
possible mechanisms of iSOA formation, in order to more accurately assess the environments in
which this pathway will operate, it remains important to further quantify (*a*) the branching
between the formation of the $C_5H_{11}O_6$ peroxy radical versus the formation of IEPOX from the
reaction between ISOPOOH and OH, (*b*) the fate of the $C_5H_{11}O_6$ peroxy radical under various
environmental conditions, as well as (*c*) the volatility of the SOA formed under various
environmental conditions and (*d*) the role of the broader suite of oxidation products in the
formation of this non-IEPOX SOA.
We present laboratory chamber studies of the gas- and particle-phase composition
resulting from both the low- and high-$NO_x$ photochemical oxidation of isoprene with the goal of
better understanding the chemical mechanisms of iSOA formation and the evolution of its
volatility and composition over time, specifically points (*c*) and (*d*) above. We compare the





observed gas-particle partitioning of several oxidation products to an assumption of equilibrium
partitioning theory. In this analysis, we use the measured thermograms of particle-phase
components to assess commonly used group-contribution methods for estimating saturation
vapor concentrations, C*. Moreover, we use a combined composition-volatility framework
[*Lopez-Hilfiker et al.*, 2015] to quantify the presence of more refractory oligomer-like
components of the SOA. We compare our laboratory results to ambient measurements taken
during the Southern Oxidant and Aerosol Study (SOAS) 2013 field campaign where similar
isoprene photochemical SOA tracers were observed. From these analyses we find (*i*) the direct
effect of higher $NO_x$ (i.e. all else being constant) is a suppression of iSOA yields at very high
input NO concentrations (50 ppb); (*ii*) a large shift to more refractory components and N-
containing products with increasing $NO_x$; (*iii*) a generally important role for accretion reactions
and other multiphase chemistry irrespective of $NO_x$ concentrations, even at relatively low
precursor concentrations, likely involving a broad suite of isoprene oxidation products.

**2**      **Experimental methods**
**2.1**      **Chamber Operation**
Experiments were performed in the Pacific Northwest National Laboratory's (PNNL) 10.6 m$^3$
polytetrafluoroethylene (PTFE) environmental chamber. The chamber has been described in
detail elsewhere [*Liu et al.*, 2012], and a portion of the data discussed herein were obtained from
the same experiments described in Liu et al. [2016]. Additional experiments with identical
chamber operation were conducted to examine a wider range of oxidant conditions. The chamber
was primarily operated in continuous-flow mode where reactants were continuously delivered at
a constant rate to allow reaction precursors and products to reach steady state concentrations



[*Shilling et al.*, 2008]. The extent of reaction is controlled by oxidant concentrations and the
residence time of air within the chamber, typically 5.2 hours. We also discuss a time-dependent
"batch mode" experiment also performed during 2015 for comparison purposes where the
chamber is filled with a fixed amount of isoprene and oxidant precursors in the dark and then the
chemistry is followed for ~6 hours after turning on the UV-VIS lights.
Isoprene was delivered into the chamber via a calibrated cylinder (Matheson, 20 ppm in
nitrogen) and mass flow controller. OH radicals were generated by the photolysis of $H_2O_2$. An
aqueous solution of $H_2O_2$ was introduced into the chamber via an automated syringe operated at
various flow rates to achieve a range of $H_2O_2$, and therefore OH and $HO_2$, concentrations.
Monodisperse, 50 nm diameter solid ammonium sulfate seed particles were continually added to
facilitate the partitioning of oxidized VOC onto particle surfaces as opposed to chamber walls
[*Zhang et al.*, 2014] for the formation of SOA. When desired, NO was added via a calibrated
cylinder (Matheson, 500 ppm in nitrogen) and mass flow controller. During the continuous-flow
experiments RH was controlled to ~50 %, while the batch mode experiment was performed
under dry conditions (~10% RH).

**2.2    Instrumentation**
A suite of online instruments were utilized to monitor gas- and particle-phase composition.
Ozone and $NO/NO_2/NO_x$ concentrations were measured using commercial instruments (Thermo
Environmental Instruments models 49C and 42C, respectively). Aerosol number and volume
concentrations were measured with a scanning mobility particle sizer (SMPS). An Aerodyne
high-resolution time-of-flight aerosol mass spectrometer (HRToF-AMS) monitored bulk



submicron organic and inorganic aerosol composition. The evolution of isoprene was monitored
with an Ionicon proton-transfer-reaction mass spectrometer (PTR-MS).
A high-resolution time-of-flight chemical ionization mass spectrometer (HRToF-CIMS)
using iodide-adduct ionization as described previously [*Lee et al.*, 2014] was coupled to a Filter
Inlet for Gases and AEROsols (FIGAERO) [*Lopez-Hilfiker et al.*, 2014] for measuring a suite of
oxygenated products in the gas- and particle-phase. The HRToF-CIMS provides measurements
of molecular composition, although cannot provide structural information and therefore cannot
differentiate between isobaric compounds. Briefly, the FIGAERO is an inlet manifold that
allowed for measurement of both gas- and particle-phase molecular composition with
approximately hourly time resolution. To collect particles, chamber air was drawn through a 1.27
cm OD (2014) or 0.635 cm OD (2015) stainless steel tube at 2.5 slpm across a Teflon filter
(Zefluor® 24 mm diameter, 2.0 μm pore size, Pall Corp.) for 31 (2014), 42 (2015), or 25 (batch)
minutes. Through a separate inlet chamber air was simultaneously sampled at 22 slpm (2014) or
12 slpm (2015) through a 1.9 cm OD, 2 m long (2014) or 1.1 m long (2015) PTFE tube for gas-
phase analysis. The gas-phase analysis required sub-sampling a portion of the flow after dilution
to maintain linearity of response in the chemical ionization. After a particle collection period,
gas-phase analysis ends and the filter containing collected particles is actuated to a location
downstream of an ultra-high purity (UHP) $N_2$ source and immediately upstream of an orifice into
the HRToF-CIMS. UHP $N_2$, continually passed across the filter at 2.5 slpm, was heated at a rate
of 10 or 15 °C min$^{-1}$ to 200 °C for a temperature-programed thermal desorption and then kept at
200 °C for the remainder of the desorption time (60 min total 2014, 70 min 2015, 40 min batch).
The coupled FIGAERO HRToF-CIMS will be referred to herein as the FIGAERO-CIMS. The
temperature axis of the FIGAERO thermograms is calibrated with compounds having known



enthalpies of sublimation [*Lopez-Hilfiker et al.*, 2014]. Lopez-Hilfiker et al. [2014] have shown
that pure compounds, or mixtures of non-interacting compounds, have consistent thermogram
shapes throughout time and reach a maximum signal at characteristic temperature ($T_{max}$) which
can be related to their enthalpies of sublimation and therefore sub-cooled pure component vapor
pressures. In this way, the $T_{max}$ of detected compounds can be used to estimate their $C^*$ at
ambient conditions even if the structure is unknown.

**2.3    Experimental Overview**
Figure 1 presents a time series of all steady-state experiments. The left and right columns contain
experiments conducted in May of 2014 and 2015, respectively. The top panels show the input
concentrations of isoprene, hydrogen peroxide, and NO, as well as the isoprene and $C_5H_{10}O_3$
(ISOPOOH + IEPOX) concentrations measured at the chamber output. The phrases "input NO",
"input $H_2O_2$", and "input isoprene", refer to the concentration of precursor that would be in the
chamber if there were no loss mechanisms except for dilution. For example, in Figure 1, top, the
input isoprene (dashed green line) is flat, while the amount of isoprene remaining in the
chamber, i.e. what is measured with the PTR-MS (solid green line), varies depending on the
chamber chemical environment. Thus, while we state that we input 0-50 ppb NO in the chamber,
in reality steady-state NO concentrations in the chamber are much lower for the majority of the
chamber residence time, in fact, usually below the detection limit of the NO analyzer due to loss
mechanisms such as nitrate formation and wall deposition.
The top row of Figure 1 portrays the time series of gas-phase species: input
concentrations of isoprene which were generally similar across both years (26 ppbv 2014, 20
ppbv 2015), NO, and $H_2O_2$, as well as gas-phase measurements of the isoprene remaining in the





chamber and $C_5H_{10}O_3$. As discussed above, the HRToF-CIMS cannot differentiate isobaric
compounds and thus $C_5H_{10}O_3$ represents the sum of ISOPOOH and IEPOX. It is important to
note that while we are suppressing the uptake of IEPOX into the particle-phase, it is still
produced at a yield of about 70-80% [*St Clair et al.*, 2016] from the reaction of ISOPOOH + OH.
The middle row shows the organic aerosol (OA) as measured by the AMS with the AMS blanks
highlighted in black squares. Steady-state periods for analysis were determined by an
unchanging OA concentration over a period of 2 or more hours typically at least 24 hours after
an intentional change in experimental conditions. All AMS data here has been multiplied by a
factor of 1.5 to correct for particle wall losses. The bottom panels show the time series of a few
dominant particle-phase components as measured by the FIGAERO-CIMS: $C_5H_{12}O_6$, $C_5H_{12}O_5$,
and $C_5H_{11}NO_7$. The organic nitrate is scaled by a factor of 20 to show its time series on the same
scale, although it is near zero when NO is not added to the chamber. The particle-phase
FIGAERO data has also been multiplied by a factor of 1.5 to correct for particle wall losses. The
grey shaded areas in the left column indicate when there was a chamber cleaning followed by a
dark $NO_3$ + isoprene experiment that is not discussed here. By systematically scanning $H_2O_2$ and
NO concentrations independently, we were able to test the response and composition of the SOA
across a range of oxidant conditions, ranging from more pristine to polluted in terms of $NO_x$
concentrations.

## 3    Results & Discussion
### 3.1    Effect of $NO_x$ on Major Gas- and Particle-Phase Species
The gas- [*Krechmer et al.*, 2015] and particle-phase [*Liu*, 2016] species detected from isoprene
photochemical oxidation when examining the non-IEPOX SOA pathway have been discussed





previously. These studies identified several $C_5H_{8-12}O_{4-8}$ compounds, among many others, and the
findings presented here are broadly consistent. Figure 2 summarizes all compounds measured as
an iodide-adduct in both the gas- (top) and particle- (bottom) phases at both low (left) and high
(right) input NO (20 ppb) for average spectra at steady state. The square root of the background
subtracted signals were taken to show the dynamic range and then normalized to the maximum
signal within each individual plot. Green bars represent organic compounds with formula
$C_xH_yO_zI^-$, while blue are organic nitrates (OrgN) with formula $C_xH_yNO_zI^-$. It is possible that
dinitrates were measured, but due to their occurrence at masses where non-nitrates would be
observed, they are difficult to conclusively identify and thus are not presented here. Major peaks
are labeled with letters corresponding to compounds in Table 1. It is important to note that while
the same molecular composition may be present in both the gas- and particle-phase, we do not
suggest that they all exist as the same structure in each phase, although some likely do. We will
discuss this further in later sections.

From Figure 2, the two largest signals detected by the FIGAERO-CIMS in the gas-phase

at both low and high input NO are $CH_2O_2$ (presumably formic acid) and $C_5H_{10}O_3$ (presumably
the sum of IEPOX and ISOPOOH). With the addition of NO, the $CH_2O_2$ signal becomes
noticeably larger than that of $C_5H_{10}O_3$, likely due to increased fragmentation. Even without
adding NO to the chamber there is still a small amount of $NO_x$ present, likely from photolysis of
inorganic nitrate on the chamber walls, as we measure non-negligible OrgN concentrations,
although the signal is very small relative to organics. The amount of OrgN in the gas-phase
increases with increased NO addition as expected. The majority of the OrgN compounds have 5
or fewer carbons and no one component dominates the OrgN. Notable signals include for
example $C_4H_7NO_5$ and $C_5H_9NO_{5-6}$. The two largest signals detected by the FIGAERO-CIMS in




the particle-phase at both low and high input NO are $C_5H_{12}O_6$ and $C_5H_{12}O_5$. Other compounds
with the isoprene $C_5$ backbone but one degree of unsaturation also represent some of the largest
signals at low-$NO_x$, such as $C_5H_{10}O_{4-7}$. As in the gas-phase, no one component dominates the
particle-phase OrgN, although one of the strongest signals is $C_5H_{11}NO_7$, the nitrate analogue to
$C_5H_{12}O_6$, formed from the same $C_5H_{11}O_6$ peroxy radical. Compounds with the formula
$C_5H_{7,9,11}NO_{4-8}$ are all observed in the particle-phase, consistent with field observations from an
isoprene-emitting forest during the SOAS campaign [*Lee et al.*, 2016]. For compounds that are
detected in the particle-phase, their $T_{max}$ and thermogram shape are also listed in Table 1 and
lends information on the nature of these compounds which will be discussed in further detail
later on.

The general effect of $NO_x$ on the SOA in this system has been described previously [*Liu*,

2016]. Here we highlight the effect of input NO concentrations on individual compounds by
focusing on three of the most prominent particle-phase species (Fig. 3, top). As the input NO
concentration increases, $C_5H_{12}O_6$ and $C_5H_{12}O_5$ decrease nonlinearly. $C_5H_{11}NO_7$, presumably
produced from the ISOPOOH + OH $C_5H_{11}O_6$ peroxy radical increases initially with increasing
NO addition. Above moderate NO input (>10 ppb), $C_5H_{11}NO_7$, a nitrate, begins to decrease with
further increases in NO addition, likely a result of ISOPOOH also decreasing as the $C_5H_9O_3$
peroxy radical reacts more with NO as opposed to $HO_2$. This behavior supports previous
observations of low isoprene SOA yields at high $NO_x$ [*Kroll et al.*, 2005; 2006; *Lane et al.*, 2008;
*Xu et al.*, 2014; *Zhang et al.*, 2011], though we note a monotonic $NO_x$ dependence of SOA yield
in our experiments. The bottom panel of Figure 3 depicts the mass fraction of OrgN as a function
of input NO. The mass fraction of OrgN increases rapidly between 0 and 10 ppb NO input and
more modestly above that. At the highest input NO concentration, OrgN make up ~40% of the





organic aerosol mass detected by the FIGAERO-CIMS. This estimate carries uncertainty due to
the inability to calibrate to every OrgN compound, as well as a lack of a single dominant OrgN.
At the highest input NO, the AMS measurements also indicate that OrgN make up ~40% of the
SOA mass, assuming a molecular weight of the typical OrgN of 148 g/mol based on the
measured FIGAERO-CIMS particle-phase OrgN distribution, which is consistent with our
findings. Though considerable uncertainties exist with respect to quantification of OrgN using
both the AMS and the FIGAERO-CIMS, the agreement between these independent
measurements suggests the calibration factors applied to the FIGAERO-CIMS OrgN signals are
reasonable. We draw two main conclusions from this analysis: (1) the complementary increase in
OrgN and decrease in non-nitrates likely accounts for the stable SOA mass yields at lower input
NO concentrations as reported previously [*Liu*, 2016], with the highest input NO concentrations
resulting in a decrease in both OrgN and non-nitrates, corresponding to the sharp decrease in
SOA mass yield at the highest input NO concentration (50 ppb), and (2) while there is no one
OrgN that is most prominent in the gas or particle phase, the total OrgN can compose up to 40%
of the SOA mass at high input NO concentrations (50 ppb).

### 3.2    Time Evolution of Low $NO_x$ Isoprene SOA Composition

To examine how isoprene photochemical SOA evolves over time, a time-dependent experiment
was conducted (Fig. 4) similar to a previous batch mode study [*Kroll et al.*, 2006]. In this "batch
mode" experiment, isoprene, $H_2O_2$, and solid ammonium sulfate seed were injected into the
chamber, and then the lights were turned on. The chemistry of the closed system was allowed to
evolve in time without further input of reactants. Each pie chart represents a FIGAERO-CIMS
particle-phase desorption measurement over the course of the experiment. The data is converted



to mass concentration as discussed previously [*Lee et al.*, 2014; *Liu*, 2016], the overall size of
each pie chart is proportional to the amount of AMS measured OA (9.8, 15.0, 14.8, 14.6 μg/m$^3$
from left to right, corrected for particle wall loss), and the time is the mid-point of the particle
collection period (which lasted 25 minutes) relative to the initiation of the chemistry. The
desorption just prior to the isoprene injection is used as the baseline, and the corresponding mass
spectra are subtracted from the succeeding desorptions. Unlike the work of Kroll et al. [2006]
who saw SOA volume maximize after ~3-4 hours of oxidation followed by a large decrease in
total volume attributed to photochemical processing, the measurements presented here did not
follow the reaction progress long after the maximum OA concentration was achieved (<1 hour)
and thus we did not observe a significant decrease in mass.

The absolute and relative concentration of $C_5H_{12}O_6$ in the particle-phase decreases from

50% of the particle-phase SOA to 25% over the four hours of oxidation. The absolute mass of
SOA also changes, primarily increasing, during the experiment, reaching a peak of 15.8 μg/m$^3$ at
t=4.3 hours. This suggests $C_5H_{12}O_6$ is transforming either within the particle-phase via
hydrolysis or other mechanisms, or in the gas-phase, with efficient gas-particle equilibration, due
to reaction with OH or photolysis [*Baasandorj et al.*, 2010; *Hsieh et al.*, 2014; *Roehl et al.*,
2007]. Gas-phase oxidation seems unlikely given that typically greater than 95% of the $C_5H_{12}O_6$
is found in the particle-phase (shown below, Fig. 5) when OA > 2 μg/m$^3$. While many of the
detected compounds are present at constant mass fractions throughout time, $C_5H_{12}O_5$, $C_5H_{12}O_4$,
and $C_5H_{10}O_3$ increase. $C_5H_{12}O_5$ has been observed previously in the gas-phase from ISOPOOH
oxidation [*Krechmer et al.*, 2015], and was also shown to be a large fraction of the particle-phase
from isoprene oxidation [*Liu*, 2016], but its production mechanism is uncertain. Krechmer et al.
[2015] suggest it could be formed from the oxidation of an impurity in the ISOPOOH, although





the experiments presented here use isoprene as the BVOC precursor, ruling out this explanation.
In these experiments, $C_5H_{12}O_5$ is observed within the first hour of isoprene oxidation and grows
to ~25% of the OA mass within 1.5 hours, becoming relatively stable thereafter. One possible
source of this compound is $RO_2 + RO_2$ reactions of the ISOPOOH derived peroxy radical and
peroxy radicals from a dihydroxy alkene reacting with $HO_2$. It is also possible that it could be
formed in the condensed phase from hydrolysis reactions. Further work is required to understand
the source of this compound.
The other two compounds that increase with time, $C_5H_{12}O_4$ and $C_5H_{10}O_3$, likely isomers
of 2-methyl tetrols and alkene triols respectively, are traditional markers of IEPOX derived SOA
[*Claeys et al.*, 2004; *Ding et al.*, 2008; *Edney et al.*, 2005; *Kourtchev et al.*, 2005; *Surratt et al.*,
2010; *Xia and Hopke*, 2006]. This result is unexpected given that using effloresced (solid)
ammonium sulfate seed at a RH below the deliquescence point (RH ~50%) together with an
SOA coating should strongly suppress the known acid catalyzed IEPOX multiphase chemistry
[*Gaston et al.*, 2014; *Lin et al.*, 2013a; *Lin et al.*, 2014; *Lin et al.*, 2012; *Lin et al.*, 2013b; *Liu et*
*al.*, 2014; *Nguyen et al.*, 2014; *Paulot et al.*, 2009b; *Surratt et al.*, 2010; *Surratt et al.*, 2006]. We
tested the uptake of an authentic IEPOX standard onto dry, crystalline ammonium sulfate seed
under conditions similar to these, though during continuous-flow mode, and found no
measurable uptake and SOA formation [*Liu*, 2016]. However, it is consistent with previous work
that found both of these tracers in the SOA when isoprene was oxidized in the absence of
deliquesced acidic seed [*Edney et al.*, 2005; *Kleindienst et al.*, 2009]. $C_5H_{12}O_4$ and $C_5H_{10}O_3$ are
less than 1% of the SOA for the first 2 hours and then gradually increase to 14% and 8% of the
SOA, respectively, after 4 hours. Interestingly, the FIGAERO thermograms for these tracers
have broad maxima at much higher $T_{max}$ than would be consistent with their elemental





composition. Lopez-Hilfiker et al. [2016b] noted two modes in the thermogram of $C_5H_{12}O_4$, one
with a $T_{max}$ as expected based on its structure and another with a higher $T_{max}$ indicating an
effectively lower volatility component thermally decomposing. The chemical mechanism leading
to the desorption of these tracers is unknown, but given that the experimental conditions strongly
suppressed the traditional acid catalyzed aqueous IEPOX chemistry, perhaps these tracers are not
solely derived from aqueous IEPOX chemistry but isoprene photochemical oxidation more
generally. In conclusion, $C_5H_{12}O_6$ condenses rapidly and initially makes up a majority of the
SOA mass, but over time its contribution decreases as other compounds such as $C_5H_{12}O_5$,
$C_5H_{12}O_4$, and $C_5H_{10}O_3$ increase. While our data suggests these compounds may be formed in the
particle phase from heterogeneous reactions, more work is required to determine their sources.

**3.3   Gas-particle Partitioning: Saturation Vapor Concentrations and Oligomeric**
**Content**
The volatility of the products generated from the non-IEPOX $C_5H_{12}O_6$ pathway [*Krechmer et al.*,
2015; *Liu*, 2016; *Riva et al.*, 2016] will be a crucial aspect of its contribution to SOA formation
and the lifetime of the resulting SOA against dilution, gas-phase oxidation, and depositional
losses. The capability of the FIGAERO to measure the concentration of individual species in
both the gas- and particle-phase allows for a direct measurement of the particle-phase fraction
($F_p$), which is the particle-phase concentration relative to the gas- and particle-phase
concentrations per volume of air. The $F_p$ can also be calculated from an assumption of
equilibrium absorptive partitioning theory first described by Pankow [1994] using equation 1,
where C* is the saturation vapor concentration ($\mu g/m^3$) of the pure substance and $C_{OA}$ is the
concentration of organic aerosol ($\mu g/m^3$).





$$F_p = \left(1 + \frac{C^*}{C_{OA}}\right)^{-1} \qquad\qquad (1)$$
Equation 1 neglects the activity coefficient and molecular weight differences for simplicity,
though any C* derived from a comparison to equation 1 would implicitly include these factors.
Calibration standards do not exist for a vast majority of compounds in SOA and therefore the C*
are largely unknown, mitigating somewhat the impact of such simplifications. Group-
contribution methods exist to estimate C*, where each functional group represents a discrete,
empirically determined contribution to the equilibrium vapor pressure of a compound [*Capouet*
*and Muller*, 2006; *Compernolle et al.*, 2011; *Nannoolal et al.*, 2008; *Pankow and Asher*, 2008].
These approaches carry substantial uncertainty for atmospheric SOA systems, in large part due to
the lack of enough standards to develop a robust parameterization. In addition, these approaches
do not directly address the potential of functional group interactions, such as intramolecular
hydrogen bonding, which when not included can lead to C* estimates that are significantly
biased low [*Kurten et al.*, 2016].

Measured $F_p$ were determined using the FIGAERO-CIMS for a subset of major particle-

phase components from 2015 (Fig. 5). These estimates include the uncertainty associated with
differences in inlet and chamber wall losses of vapors relative to particles. Operating the
chamber in continuous flow mode likely reduces the impact of chamber walls, at least for low
volatility to semi volatile compounds, as some degree of equilibration can occur [*Liu*, 2016;
*Shilling et al.*, 2008]. A short (1-2 m) laminar flow Teflon inlet line with a short residence time
(<1 s) was coupled to the chamber for the detection of gases by the FIGAERO-CIMS. The
diffusion-controlled loss of gases in the tubing is likely less than 50% at most, again a small
effect on the comparison of measured and predicted $F_p$ as we will show below.





$F_p$ were predicted using equation 1 with C* calculated via the EVAPORATION group-
contribution method [*Compernolle et al.*, 2011], which generally gave similar estimates as the
Capouet and Muller approach [2006]. The Nannoolal method [2008] was also explored, but it
gave C* estimates that varied by several orders of magnitude for structurally similar compounds,
as well as estimates that were unexpectedly low based on FIGAERO measurements and what
one would expect based on molecular structure, consistent with previous findings [*Kurten et al.*,
2016]. The SIMPOL method of Pankow and Asher [2008] was also applied to select compounds
and is discussed below. A major limitation of this analysis is that we do not know the structure of
the molecules detected, only the elemental composition, and so we make assumptions based on
the most likely functional groups expected from the chemical conditions of the chamber and
from the elemental composition (e.g., degrees of unsaturation, oxygen to carbon ratio). In many
cases these assumptions have little impact on our conclusions.
If the SOA formed according to equilibrium partitioning theory as first described by
Pankow [1994], the $F_p$ measured by the FIGAERO and the C* calculated using group-
contribution methods should be in agreement over a range of organic aerosol mass
concentrations. Figure 5 indicates two immediate challenges to testing partitioning theory. First,
a large number of mostly small carbon number compounds have a much higher measured $F_p$
relative to the predicted $F_p$ based on their group-contribution determined C*. The C* estimates
would have to be in error by at least five or more orders of magnitude, which is likely not the
case as there are many measurements of vapor pressures for similar compounds. Furthermore,
the thermograms of these compounds appear broad, not Gaussian as one would expect for
individual non-interacting compounds [*Lopez-Hilfiker et al.*, 2014], and do not peak until ~85 °C
or higher, see Table 1, which is also inconsistent with the calibrated composition-enthalpy of



sublimation relationship scaled for the FIGAERO used here [*Lopez-Hilfiker et al.*, 2014]. We
attribute this behavior to thermal decomposition of lower volatility components during the
desorption process giving rise to smaller molecular weight, more volatile components as in
previous studies of IEPOX SOA tracers in the southeast U.S. [*Lopez-Hilfiker et al.*, 2016b], α-
pinene derived chamber SOA [*Lopez-Hilfiker et al.*, 2015], and biomass burning organic aerosol
in the northwest U.S. [*Gaston et al.*, 2016]. That is, the disagreement between measured and
predicted $F_p$ for these compounds is not necessarily a failure of equilibrium partitioning theory,
nor evidence that equilibrium had not been achieved, but rather that the compounds desorbing
were actually part of another larger molecular weight accretion product, the C* and gas-phase
concentrations of which are unknown.

The second challenge to testing gas-particle partitioning is illustrated in the $F_p$ for two

representative compounds, $C_5H_{12}O_6$ and $C_5H_{10}O_6$ (Fig. 5, bottom panels). That there is
reasonable agreement between measured and predicted $F_p$ (Fig. 5, bottom right, circles) for
$C_5H_{10}O_6$ suggests that equilibrium partitioning is potentially achieved in the chamber. However,
that $C_5H_{12}O_6$ is not in good agreement (Fig. 5, bottom left, circles) cannot be explained by
thermal decomposition of lower volatility material, because the thermogram shape is Gaussian
and therefore looks like a pure component and the predicted $F_p$ is much larger than measured
(opposite to the above situation). This behavior implies inaccurate C* derived from the group-
contribution methods.

The EVAPORATION group-contribution method [*Compernolle et al.*, 2011] used in

Figure 5 and that of Capouet and Muller [2006] both produce a C* of 0.03 µg/m$^3$ for $C_5H_{12}O_6$
when assuming it is a dihydroxy dihydroperoxide, while the SIMPOL method predicts 2 µg/m$^3$
[*Pankow and Asher*, 2008]. For $C_5H_{10}O_6$, assumed to be a hydroxy dihydroperoxy aldehyde,





EVAPORATION [*Compernolle et al.*, 2011] predicts 4 µg/m$^3$, the Capouet and Muller [2006]
method predicts 2 µg/m$^3$, and SIMPOL [*Pankow and Asher*, 2008] suggests 15 µg/m$^3$. That the
$C_5H_{12}O_6$ values vary by 2 orders of magnitude while the $C_5H_{10}O_6$ values vary by a factor of 7
suggests the need for better experimental constraints. Using the measured $T_{max}$ from the
FIGAERO thermograms, the $C^*$ for $C_5H_{12}O_6$ and $C_5H_{10}O_6$ were determined to be 0.7 and 6.7
µg/m$^3$, respectively. These values were then used to re-calculate the predicted $F_p$ using equation
1. The original group-contribution calculated $F_p$ is shown alongside the adjusted points in the
bottom panels of Figure 5 as navy crosses. The root mean square error of both $C_5H_{12}O_6$ and
$C_5H_{10}O_6$ is minimized when comparing the measurements with the adjusted $F_p$, indicating that
the calibrated FIGAERO temperature axis can more accurately determine the $C^*$.

In the case of $C_5H_{12}O_6$ the FIGAERO determined $C^*$ is much closer to the SIMPOL

estimation, and significantly higher than that estimated by the other group-contribution methods.
We suspect the large differences between measured and group-contribution method estimates of
$C^*$ in this case are due to the lack of vapor pressure data on compounds with hydroperoxide
groups, specifically multifunction hydroperoxides: there is only data on four hydroperoxide
containing compounds on which these models are based [*Capouet and Muller*, 2006;
*Compernolle et al.*, 2011]. It can be argued that -OH and –OOH groups will lower the $C^*$
relative to the precursor by a comparable amount, with the possibility that a -OOH group could
lower it slightly more due to its additional oxygen. For example, the $C^*$ of 2-methyl-1,2,3,4-
butanol ($C_5H_{12}O_4$) was calculated to be 9 to 34 µg/m$^3$, depending on the method used [*Capouet*
*and Muller*, 2006; *Compernolle et al.*, 2011], roughly 250-1000 times greater than the $C^*$
estimates for $C_5H_{12}O_6$. That is, a molecule with the same number of distinct –OH containing
functional groups is predicted to have a very different $C^*$ because the vapor pressure lowering of





–OOH groups assumed by these group-contribution methods is larger than that for an -OH
group. Our observations suggest this assumption is likely incorrect, at least to the extent
employed in these methods, and is supported by previous work that found group-contribution
methods predicted significantly lower C* than models did for compounds with multiple
functionalities [*Kurten et al.*, 2016; *Valorso et al.*, 2011]. Conversely, we assume $C_5H_{10}O_6$ is a
hydroxy dihydroperoxy aldehyde, also with two –OOH groups, but the group contribution
methods accurately predicte a C* consistent with that inferred from the FIGAERO $T_{max}$. This
agreement may be a coincidence or indicative of the fact that multifunctional compounds with
slightly different function groups can have significantly different intramolecular interactions,
leading to significantly different saturation vapor concentrations [*Compernolle et al.*, 2011].
There are two main conclusions we draw from Figure 5. The first is that testing
equilibrium partitioning theory is a challenge without a direct constraint on the C* like the
FIGAERO desorption $T_{max}$ due to possibly large systematic errors in the C* predicted from
group-contribution methods. Moreover, thermal decomposition of higher molecular weight
compounds, such as oligomers, into smaller molecular weight compounds generates uncertainty
in the measured $F_p$ in that the FIGAERO $T_{max}$ derived C* does not correspond in such cases to
the observed molecule. From this result we draw our second conclusion: a surprisingly large
fraction of the iSOA is resistant to evaporation, indicating it will have a longer lifetime against
dilution [*Kroll et al.*, 2006]. Approximately 30-45% of the SOA mass detected by the
FIAGERO-CIMS desorbs at temperatures greater than 80 °C, much of that above 100 °C, which
corresponds to effective enthalpies of sublimation >150 kJ/mol in our FIGAERO assuming no
diffusion limitations to evaporation from the particles that exist at these temperatures (for
example, due to highly viscous phases). We note that Kroll et al. [2006] also found evidence for



a significant mass fraction of large molecular weight compounds when applying the AMS to
low-$NO_x$, non-IEPOX iSOA. We conclude that accretion products are the cause of this more
refractory SOA component, but, we cannot determine from the thermograms alone whether the
accretion process is reversible at ambient temperatures on longer timescales than the ~1 hour
desorptions. That the SOA yield from isoprene is significantly higher for similar organic mass
loadings than that reported from ISOPOOH only, suggests an important role for the broader
distribution of oxidation products formed in addition to those from ISOPOOH. One possible
reason is that these mostly semi volatile products can contribute to lower volatility products via
accretion chemistry [*Jathar et al.*, 2016; *Sato et al.*, 2011; *Tsai et al.*, 2015].

**3.4    Role of $NO_x$ in iSOA volatility**
Previous studies using thermal denuders and either an AMS or a tandem differential mobility
measurement of particle size distributions have found that iSOA formed in the presence of $NO_x$
is less volatile relative to that formed in $HO_2$-dominant regimes [*Kleindienst et al.*, 2009; *Xu et
al.*, 2014]. We compare the iSOA volatility under different regimes by summing the FIGAERO
thermogram signals across all ions having formulae $C_xH_yO_zN_{0-1}I^-$. These sum thermograms are
plotted as a function of temperature for both low- and high-$NO_x$ conditions in Figure 6, bottom
left and right, respectively. Since $C_5H_{12}O_6$ is a large portion of the SOA mass concentration in
these experiments (e.g. Fig. 4), it is shown separately in dark green with the remainder of the
summed signal shown in light green. $C_5H_{12}O_6$ is clearly a large contribution to the sum signal in
both the low- and high-$NO_x$ cases, although more so in the low $NO_x$ regime, with the remaining
thermogram signal primarily located in the lower volatility (higher temperature) "tail" of the
desorption. At high $NO_x$, there are two clear modes in the thermogram remaining after removing



the $C_5H_{12}O_6$ contribution (light green), one mode at roughly the same $T_{max}$, and therefore
volatility, of $C_5H_{12}O_6$, and the other mode at a higher $T_{max}$, ~110 °C, suggesting a larger fraction
of detected iSOA mass at high $NO_x$ is resistant to evaporation compared to the low $NO_x$ case.
We also note that the $T_{max}$ of individual compounds shifts to higher values with the addition of
$NO_x$ except for the highest mass compounds (see Table 1). Although the thermograms for many
of these compounds do not have a distinct Gaussian shape, making determination of the $T_{max}$
uncertain or undefined, the shift to higher $T_{max}$ for the same compounds could indicate not just
lower volatility products in the form of oligomers, but also potentially a change in the overall
particle viscosity causing the iSOA to be more resistant to evaporation with the addition of $NO_x$.

A sum thermogram of α-pinene ozonolysis that has been previously reported [*Lopez-*

*Hilfiker et al.*, 2015] is displayed alongside those of the low- and high-$NO_x$ experiments (Fig. 6,
top) for comparison. The α-pinene SOA has a bimodal sum thermogram, similar to that of the
high-$NO_x$ iSOA with the second lower volatility modes in the same temperature range. The
higher volatility mode of the α-pinene SOA corresponds in temperature space to that of the
higher volatility mode of the low-$NO_x$ iSOA. The multiple modes of the α-pinene sum
thermogram have the same relative maxima in signal space, unlike the isoprene sum
thermograms. α-Pinene ozonlysis apparently generates a larger fraction of lower volatility SOA
relative to isoprene photochemical oxidation, although isoprene photochemical SOA has
components in the same volatility ranges of α-pinene ozonlysis SOA, and the relative size of the
various modes and location in temperature space is dependent on the amount of $NO_x$.

It is important to note that the contribution of the effectively lower volatility components

inferred from thermograms in Figure 6 is likely underestimated in both the low- and high- $NO_x$
cases because the thermograms are presented as ion signal space, not mass concentration. If we





converted into mass concentration units prior to calculating the summed thermogram, the
contribution of $C_5H_{12}O_6$ would be significantly less than implied in Figure 6. The integrated
contribution of $C_5H_{12}O_6$ would instead be more similar to that shown in Figure 4 for which we
applied calibration estimates based on ISOPOOH and a range of other oxygenated compounds,
together with the ion-molecule collision limited sensitivity discussed previously [*Liu*, 2016;
*Lopez-Hilfiker et al.*, 2016a]. In conclusion, the low-$NO_x$ SOA has an overall higher volatility
and the addition of $NO_x$ results in lower volatility material making up a larger fraction of the
SOA, although the total SOA yield is lower [*Liu*, 2016], in general agreement with previous
studies showing that increasing $NO_x$ leads to lower volatility SOA [*Kleindienst et al.*, 2009],
likely by enhancing oligomerization [*Dommen et al.*, 2006; *Xu et al.*, 2014]. However, many of
these previous studies were carried out in very different concentration regimes with different
detection techniques, so that the data we present offers an additional contribution to the general
importance of oligomerization.

**3.5     Comparisons to Ambient SOA Formed in an Isoprene-Rich Environment**
During the summer of 2013, the Southern Oxidant and Aerosol Study (SOAS) was
conducted in Brent, AL, as described previously [*Attwood et al.*, 2014; *Lee et al.*, 2016;
*Washenfelder et al.*, 2015]. The same instrument was used here and for the PNNL chamber
studies described herein, allowing for a direct comparison between chamber and field
measurements. Due to the influence of anthropogenic sulfur emissions and the high relative
humidity, IEPOX multiphase chemistry contributed significantly to SOA in this region as
expected [*Budisulistiorini et al.*, 2015; *Hu et al.*, 2015; *Lopez-Hilfiker et al.*, 2016b]. However,
we also detected in the ambient SOA the dominant components of the chamber generated SOA



described above, namely $C_5H_{12}O_6$, $C_5H_{12}O_5$, $C_5H_{11}NO_7$, $C_5H_{10}O_6$, and $C_5H_{10}O_5$, the individual
diurnal profiles and the sum of all 5 are shown in Figure 7, bottom. While the FIGAERO-CIMS
provides molecular composition and not structural information, that the $C_5H_{12}O_6$ thermograms at
SOAS and PNNL look nearly identical and have the same $T_{max}$ (Fig. 7, top) provides some
support that the same compounds are present in both systems.
The mean diurnal cycle of the sum of the $C_5$ tracers detected in the PNNL chamber
exhibit a daytime maximum (Fig. 7) as expected given the strong daily modulation in isoprene
emissions [*Davison et al.*, 2009; *de Arellano et al.*, 2011; *Fuentes et al.*, 1999; *Holzinger et al.*,
2002; *Kalogridis et al.*, 2014; *Lee and Wang*, 2006; *Rinne et al.*, 2002; *Yang et al.*, 2005]. The
sum of the 5 tracers, $C_5H_{12}O_6$, $C_5H_{12}O_5$, $C_5H_{11}NO_7$, $C_5H_{10}O_6$, and $C_5H_{10}O_5$, reaches a minimum
of 40 ng/m$^3$ during dark hours and a maximum of 90 ng/m$^3$ during the day (Fig. 7). While similar
compounds were measured during SOAS and the PNNL isoprene oxidation experiments,
comparisons with field data provides interesting insight into the production of $C_5H_{12}O_5$. Figure 4
indicates that the $C_5H_{12}O_5$ SOA mass fraction increases with time relative to $C_5H_{12}O_6$, but
saturates at ~20%, and the concentration of $C_5H_{12}O_6$ is always greater than $C_5H_{12}O_5$ in the
chamber experiments. In the SOAS field data however, the relationship is reversed: the $C_5H_{12}O_5$
is always greater than the $C_5H_{12}O_6$. Although the two are correlated in the chamber ($R^2 = 0.6$),
they are not correlated in the atmosphere ($R^2 = 0.1$). While we do not have a definitive
explanation, this behavior may be due to a variety of factors. As noted above, the $C_5H_{12}O_5$ could
be produced by gas-phase $RO_2 + RO_2$ chemistry, however, we expect this pathway is more
important in the chamber with higher $RO_2$ concentrations than in the atmosphere. While also
possibly tied generally to aging as shown in Figure 4, it could be that multiphase processes such
as hydrolysis of hydroperoxides or organosulfate formation and subsequent hydrolysis are more





significant in ambient particles. That is, assuming $RO_2 + RO_2$ chemistry is less efficient in the
ambient atmosphere, a reasonable explanation is that $C_5H_{12}O_5$ represents a hydrolysis product of
the dihydroxy dihydroperoxide, $C_5H_{12}O_6$, or related organosulfates, as suggested in the recent
work of Riva et al. [2016] and that these processes are enhanced in the aqueous aerosols likely
present in the SE U.S.

**4        Conclusions**
We have explored the composition and volatility of isoprene SOA at low and high $NO_x$
concentrations utilizing effloresced ammonium sulfate seed to prevent IEPOX uptake and thus
suppress IEPOX multiphase chemistry. We measured compositions of products reported in
previous works of similar experiments [*Krechmer et al.*, 2015; *Liu*, 2016], in particular $C_5H_{12}O_6$
and related highly oxidized compounds. We examined the saturation vapor concentrations of
several of the most dominant particle-phase signals and tested the accuracy of various group-
contribution methods to determine the C*. Of the three group-contribution methods assessed, the
SIMPOL approach [*Pankow and Asher*, 2008] gave the closest estimates of C* compared to
those determined from the thermograms. The vapor pressure lowering effect of –OOH groups,
assumed to be abundantly present in this system, appears to be greatly overestimated in two
commonly used methods [*Capouet and Muller*, 2006; *Compernolle et al.*, 2011]. Through these
analyses we found that a significant fraction of SOA components we measure are likely thermal
decomposition fragments, characterized by broad thermograms and higher than expected $T_{max}$.

That such a large fraction (30-45%) of the non-IEPOX iSOA is of low volatility implies

the lifetime of non-IEPOX iSOA is longer than would previously be expected. Our findings also
suggest that experiments which assess the SOA formation potential of isoprene likely



underestimate the overall potential due to the participation of a broad suite of products in
accretion chemistry. Further work on the role of accretion chemistry in this system is needed to
verify that the higher iSOA yield observed by Liu et al. [2016] from isoprene is indeed caused by
semi volatile products participating in accretion reactions. Furthermore, we have shown here that
the addition of NO has a strong effect on the amount of $C_5H_{12}O_6$ produced, and while the overall
volatility of the OA decreases with $NO_x$, the total amount of OA also decreases [*Liu*, 2016],
indicating that in polluted regions the amount of SOA formed from this pathway will be
diminished, but the SOA will be longer lived against dilution. In conclusion, due to the high
yield of IEPOX from ISOPOOH + OH it has been assumed to be the most important pathway for
the formation of iSOA, however, its relatively high volatility (1,1691 $\mu g/m^3$ [*Compernolle et al.*,
2011]) and the fact that it requires such specific conditions to form SOA efficiently implies that
the formation of SOA from the non-IEPOX pathway discussed herein can also play an important
role in many environments regardless of sulfate aerosol concentrations.

**Acknowledgements**
This work was supported by the U.S. Department of Energy ASR grants DE-SC0011791.
E.L.D was supported by the National Science Foundation Graduate Research Fellowship under
Grant No. DGE-1256082. B.H.L was supported by the National Oceanic and Atmospheric
Administration (NOAA) Climate and Global Change Postdoctoral Fellowship Program. PNNL
authors were supported by the U.S. Department of Energy, Office of Biological and
Environmental Research, as part of the ASR program. Pacific Northwest National Laboratory is
operated for DOE by Battelle Memorial Institute under contract DE-AC05-76RL01830. We



thank the Southern Oxidant and Aerosol Study (SOAS) science team, especially J.L. Jimenez
(CU Boulder) and A.G. Carlton (Rutgers) for facilitating our participation.





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




## Figure and Table Captions

**Table 1.** Gas- and particle-phase compounds detailed in the mass spectra in Figure 2 (top). Many molecular compositions are observed in both the gas- and particle-phase. If the composition is observed in the particle phase, a $T_{max}$ is listed at both low (0 ppb input NO) and high (20 ppb input NO) $NO_x$. The desorption shape is also listed and is consistent across $NO_x$ conditions. The significance of the $T_{max}$ and desorption shape are discussed in detail in the text. If the compound is only detected in the gas phase, "NA" is listed in the $T_{max}$ and thermogram columns, indicating that those values are not applicable.

**Figure 1.** Overview of the 2014 and 2015 measurements taken at PNNL. The left column is data from the 2014 campaign, the right column is 2015. The top row shows gas-phase compounds measured by the PTR-MS and FIGAERO-CIMS, as well as input concentrations of $H_2O_2$, NO, and isoprene. Middle row shows the OA as measured by the AMS. Steady state periods are shown within magenta circles, AMS blanks as black squares. Select particle phase species measured by the FIGAERO-CIMS are in the bottom row. Grey shaded areas in each column indicate when chamber lights were off for chamber cleaning and a dark $NO_3$ experiment (in 2014) which is not discussed here. Note that the axis limits are not the same due to a wide range in concentrations across years, while $C_5H_{12}O_5$ has been enhanced 5x and $C_5H_{11}NO_7$ has been enhanced 20x in the bottom rows to clearly show the behavior of each species on the same axis.

**Figure 2.** Mass spectra for compounds with composition $C_xH_yO_zI^-$ (green) and $C_xH_yNO_zI^-$ (blue) at low (left) and high (right) NO input in both the gas- (top) and particle- (bottom) phases. Bars are sized by the square root of signal (counts s$^{-1}$ for the gas-phase, counts for the particle-phase) to show the dynamic range. Major components are labeled with letters corresponding to those found in Table 1.

**Figure 3.** Top: Normalized signals of $C_5H_{12}O_6$ and $C_5H_{11}NO_7$, believed to originate in the gas phase from the same $C_5H_{11}O_6$ peroxy radical, as well as $C_5H_{12}O_5$, as a function of input NO. Signal is normalized to maximum signal for each compound to show the relative behaviors. Bottom: The mass fraction of organic nitrates as a function of NO. Mass fraction refers to the mass concentration of FIGAERO-CIMS measured OrgN relative to the total mass concentration of organics (non-nitrogen containing + OrgN) measured by the FIGAERO-CIMS.

**Figure 4.** Time evolution of particle-phase concentrations in a batch mode isoprene photochemical oxidation experiment at low-$NO_x$. Time increases from left to right and the size of the pies is proportional to the amount of OA present which is: 9.8, 15.0, 14.8, 14.6 µg/m$^3$ from left to right.

**Figure 5.** Top: Predicted versus measured fraction in the particle-phase ($F_p$). Predicted $F_p$ is obtained from equation 1 where C*s were calculated with the EVAPORATION group-contribution method [*Compernolle et al.*, 2011] labeled as "Group-Cont. C*" in the bottom panels. Measured $F_p$ is the direct measurement from the FIGAERO. Bottom: The $F_p$ can also be predicted based on the calibrated FIGAERO temperature axis as discussed in the methods and is shown as the predicted $F_p$ here. Agreement can be reached for two representative compounds where the $F_p$ is over and correctly predicted (left, right respectively).





**Figure 6.** Top: Sum thermograms of α-pinene + $O_3$ compared to isoprene ($C_5H_8$) photooxidation with and without $NO_x$. The α-pinene sum thermogram has been reported previously (Lopez-Hilfiker et al. [2015], Fig 5). Bottom: The sum thermograms at low (left) and high (right) input NO. The thermogram of $C_5H_{12}O_6$, the largest signal in both cases, is separated out (dark green) and the sum of the remaining signal minus $C_5H_{12}O_6$ is displayed as the remaining signal (light green). At high NO input the sum of OrgN and the sum of non-nitrate organics are plotted (dashed lines, independent of solid lines) to show the relative thermogram features.

**Figure 7.** Top: Thermograms of $C_5H_{12}O_6$ observed during chamber experiments at PNNL (diamonds) compared to those measured during the SOAS field campaign (solid line). Bottom: Diurnal behavior of prominent $C_5$ compounds measured in the PNNL chamber and detected at SOAS: $C_5H_{12}O_6$, $C_5H_{12}O_5$, $C_5H_{10}O_6$, $C_5H_{10}O_5$, $C_5H_{11}NO_7$, and the sum of all 5. Arrows indicate which y-axis each compound is plotted on. The diurnal profile of these compounds maximizes during daylight hours and minimizes during the night as would be expected.



| Letter | Mass | Molecular Composition | Particle-phase $T_{max}$, Low $NO_x$ | Particle-phase $T_{max}$, High $NO_x$ | thermogram shape |
|--------|------|-----------------------|--------------------------------------|---------------------------------------|------------------|
| a | 172.9105 | $CH_2O_2I^-$ | 87 | 94 | broad |
| b | 202.9211 | $C_2H_4O_3I^-$ | 113 | 115 | broad |
| c | 216.9367 | $C_3H_6O_3I^-$ | 76 | 100 | broad |
| d | 230.9524 | $C_4H_8O_3I^-$ | 87 | 115 | broad |
| e | 244.968 | $C_5H_{10}O_3I^-$ | NA | NA | NA |
| f | 258.9473 | $C_5H_8O_4I^-$ | 76, 111 | 70, 115 | double |
| g | 275.9374 | $C_4H_7NO_5I^-$ | 88 | 115 | broad |
| h | 289.9531 | $C_5H_9NO_5I^-$ | NA | NA | NA |
| i | 305.948 | $C_5H_9NO_6I^-$ | NA | NA | NA |
| j | 246.9473 | $C_4H_8O_4I^-$ | 95 | 110 | broad |
| k | 278.9735 | $C_5H_{12}O_5I^-$ | 60 | 48 | Gaussian |
| l | 294.9684 | $C_5H_{12}O_6I^-$ | 63 | 56 | Gaussian |
| m | 323.9586 | $C_5H_{11}NO_7I^-$ | 72 | 50 | Gaussian |
| n | 339.9535 | $C_5H_{11}NO_8I^-$ | 53 | 45 | Gaussian |


**Table 1.** Gas- and particle-phase compounds detailed in the mass spectra in Figure 2 (top). Many
molecular compositions are observed in both the gas- and particle-phase. If the composition is
observed in the particle phase, a $T_{max}$ is listed at both low (0 ppb input NO) and high (20 ppb
input NO) $NO_x$. The desorption shape is also listed and is consistent across $NO_x$ conditions. The
significance of the $T_{max}$ and desorption shape are discussed in detail in the text. If the compound
is only detected in the gas phase, "NA" is listed in the $T_{max}$ and thermogram columns, indicating
that those values are not applicable.




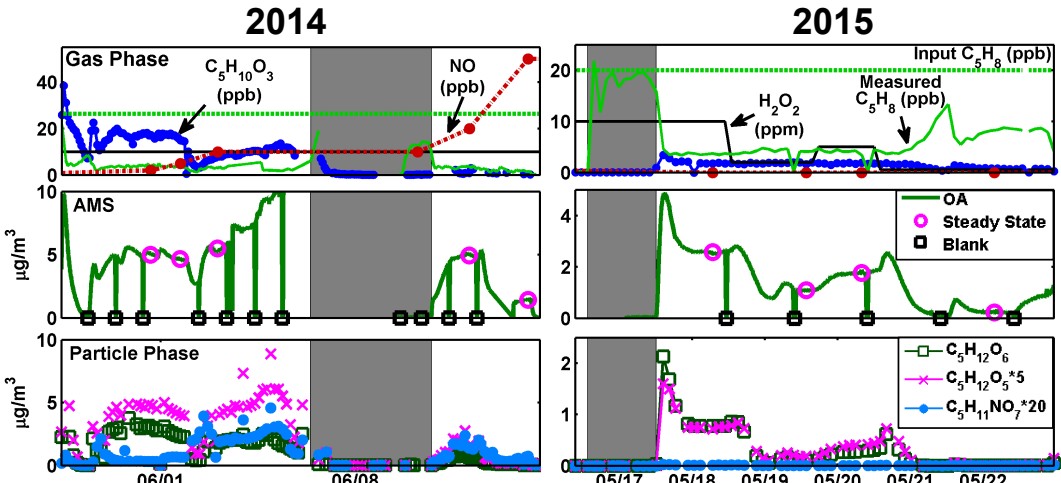

**Figure 1.** Overview of the 2014 and 2015 measurements taken at PNNL. The left column is data from the 2014 campaign, the right column is 2015. The top row shows gas-phase compounds measured by the PTR-MS and FIGAERO-CIMS, as well as input concentrations of $H_2O_2$, NO, and isoprene. Middle row shows the OA as measured by the AMS. Steady state periods are shown within magenta circles, AMS blanks as black squares. Select particle phase species measured by the FIGAERO-CIMS are in the bottom row. Grey shaded areas in each column indicate when chamber lights were off for chamber cleaning and a dark $NO_3$ experiment (in 2014) which is not discussed here. Note that the axis limits are not the same due to a wide range in concentrations across years, while $C_5H_{12}O_5$ has been enhanced 5x and $C_5H_{11}NO_7$ has been enhanced 20x in the bottom rows to clearly show the behavior of each species on the same axis.





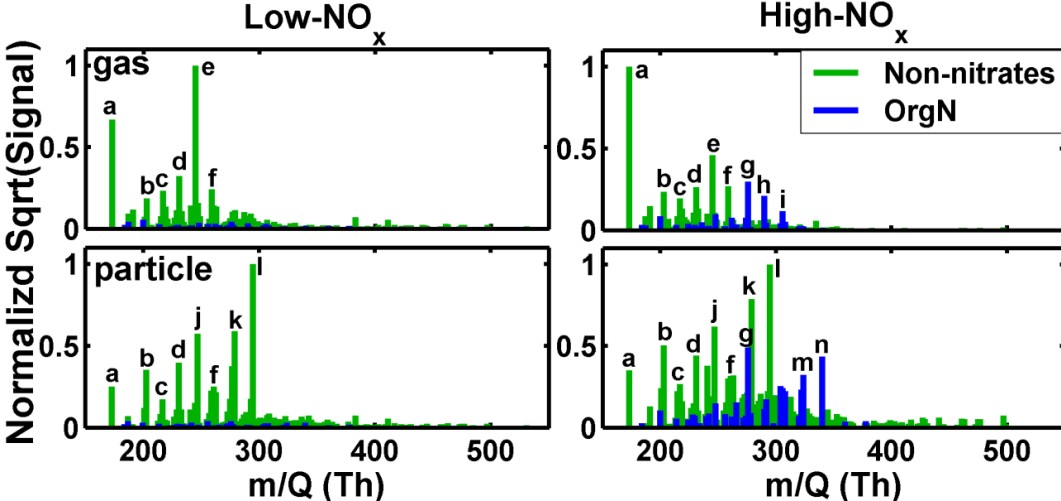

1120
1121
**Figure 2.** Mass spectra for compounds with composition $C_xH_yO_zI^-$ (green) and $C_xH_yNO_zI^-$ (blue) at low (left) and high (right) NO input in both the gas- (top) and particle- (bottom) phases. Bars are sized by the square root of signal (counts s$^{-1}$ for the gas-phase, counts for the particle-phase) to show the dynamic range. Major components are labeled with letters corresponding to those found in Table 1.





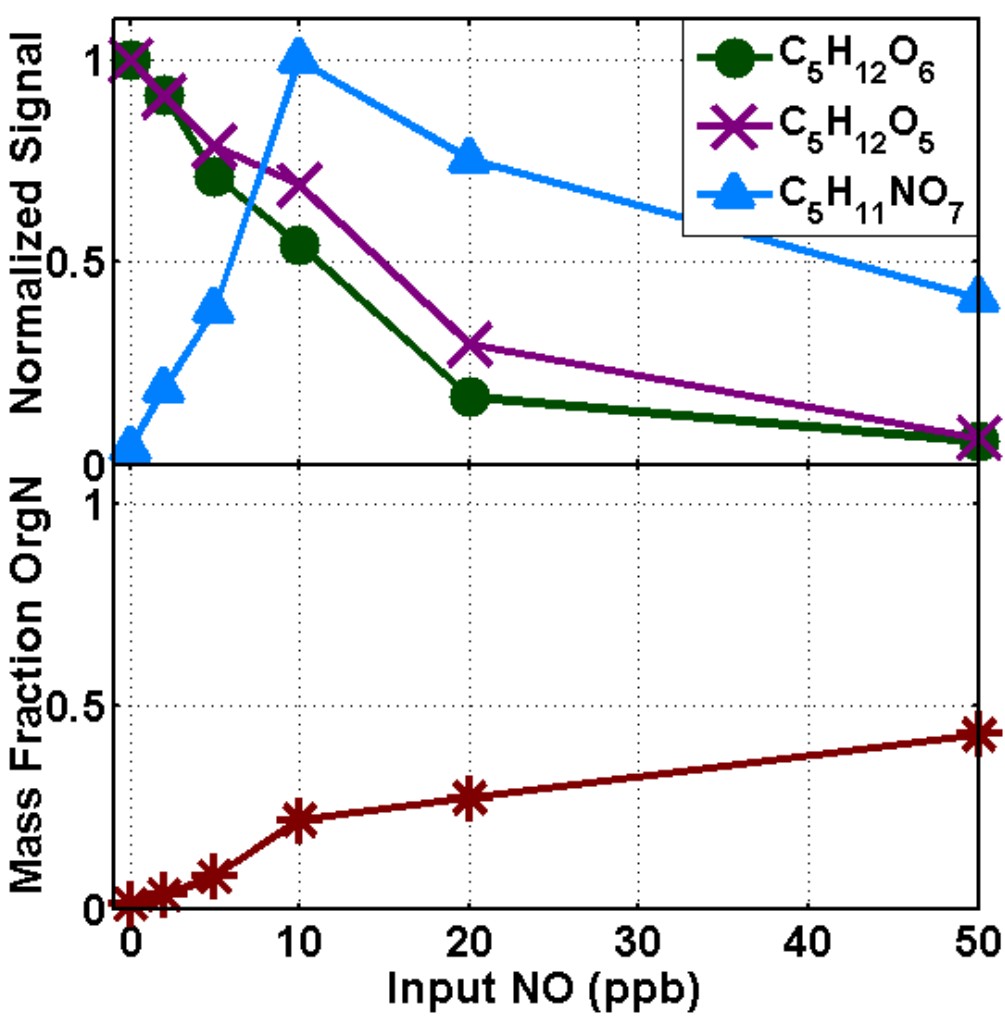

**Figure 3.** Top: Normalized signals of $C_5H_{12}O_6$ and $C_5H_{11}NO_7$, believed to originate in the gas phase from the same $C_5H_{11}O_6$ peroxy radical, as well as $C_5H_{12}O_5$, as a function of input NO. Signal is normalized to maximum signal for each compound to show the relative behaviors. Bottom: The mass fraction of organic nitrates as a function of NO. Mass fraction refers to the mass concentration of FIGAERO-CIMS measured OrgN relative to the total mass concentration of organics (non-nitrogen containing + OrgN) measured by the FIGAERO-CIMS.

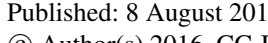





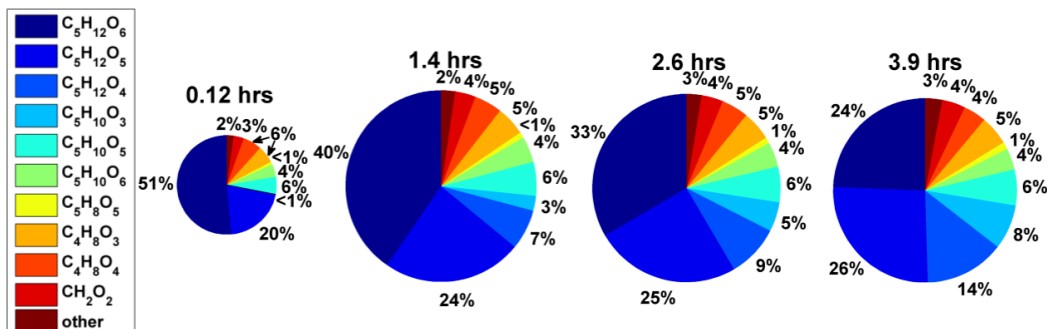

**Figure 4.** Time evolution of particle-phase concentrations in a batch mode isoprene
photochemical oxidation experiment at low-$NO_x$. Time increases from left to right and the size
of the pies is proportional to the amount of OA present which is: 9.8, 15.0, 14.8, 14.6 µg/m$^3$ from
left to right.





**Figure 5.** Top: Predicted versus measured fraction in the particle-phase ($F_p$). Predicted $F_p$ is obtained from equation 1 where $C*$s were calculated with the EVAPORATION group-contribution method [*Compernolle et al.*, 2011] labeled as "Group-Cont. $C*$" in the bottom panels. Measured $F_p$ is the direct measurement from the FIGAERO. Bottom: The $F_p$ can also be predicted based on the calibrated FIGAERO temperature axis as discussed in the methods and is shown as the predicted $F_p$ here. Agreement can be reached for two representative compounds where the $F_p$ is over and correctly predicted (left, right respectively).





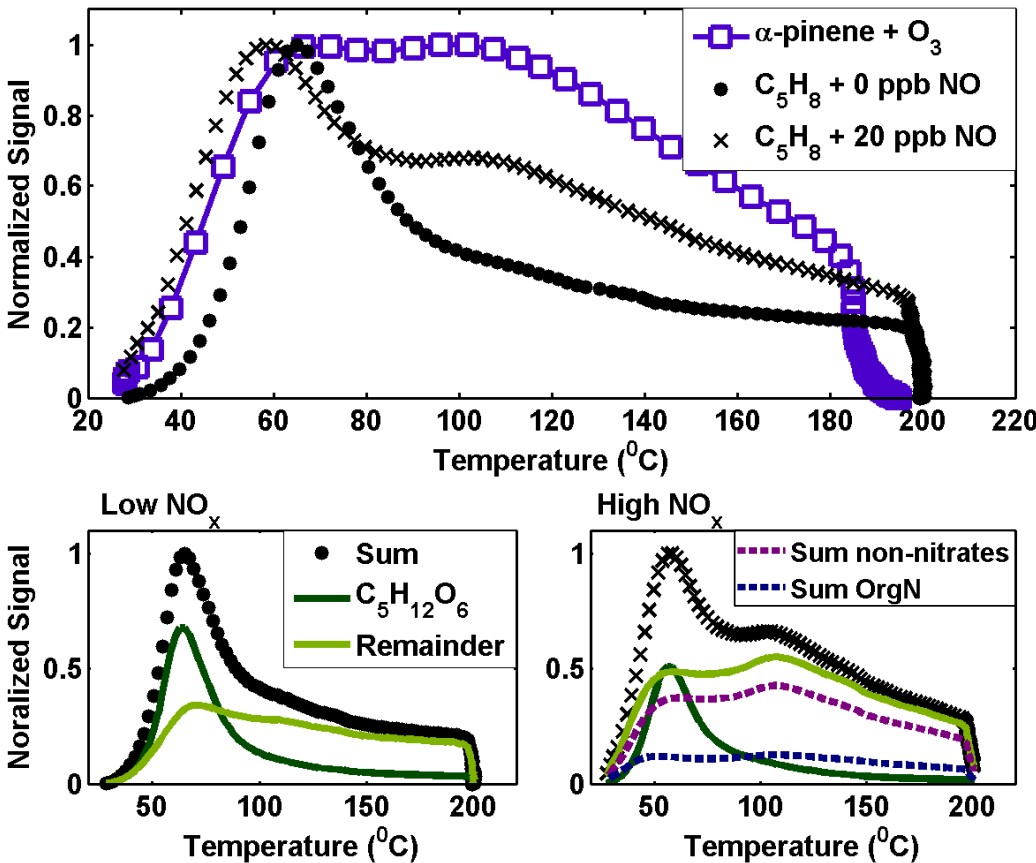

**Figure 6.** Top: Sum thermograms of α-pinene + $O_3$ compared to isoprene ($C_5H_8$) photooxidation with and without $NO_x$. The α-pinene sum thermogram has been reported previously (Lopez-Hilfiker et al. [2015], Fig 5). Bottom: The sum thermograms at low (left) and high (right) input NO. The thermogram of $C_5H_{12}O_6$, the largest signal in both cases, is separated out (dark green) and the sum of the remaining signal minus $C_5H_{12}O_6$ is displayed as the remaining signal (light green). At high NO input the sum of OrgN and the sum of non-nitrate organics are plotted (dashed lines, independent of solid lines) to show the relative thermogram features.





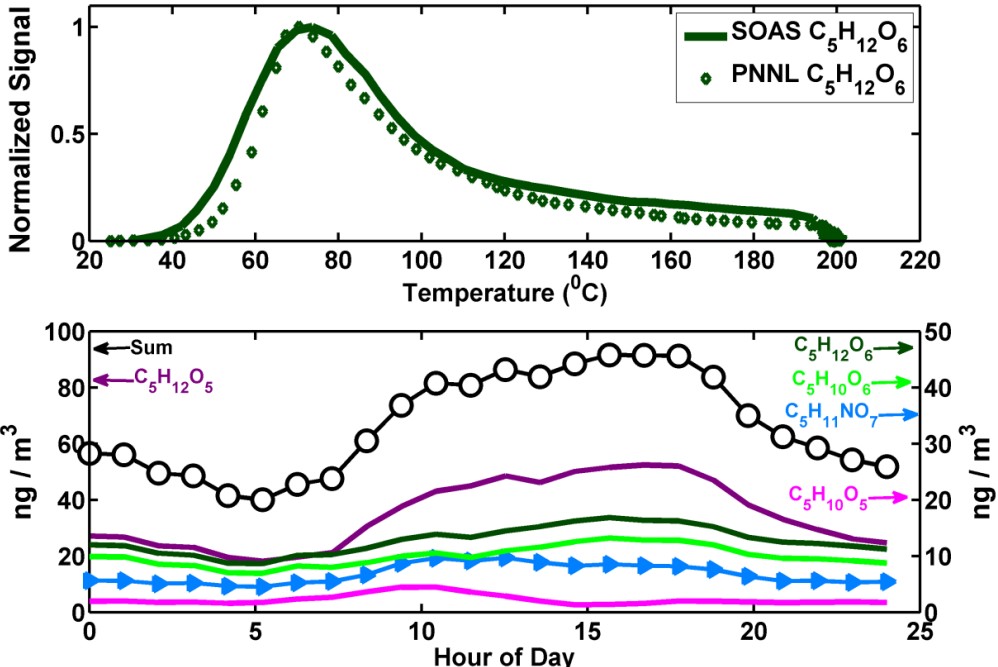

**Figure 7.** Top: Thermograms of $C_5H_{12}O_6$ observed during chamber experiments at PNNL
(diamonds) compared to those measured during the SOAS field campaign (solid line). Bottom:
Diurnal behavior of prominent $C_5$ compounds measured in the PNNL chamber and detected at
SOAS: $C_5H_{12}O_6$, $C_5H_{12}O_5$, $C_5H_{10}O_6$, $C_5H_{10}O_5$, $C_5H_{11}NO_7$, and the sum of all 5. Arrows indicate
which y-axis each compound is plotted on. The diurnal profile of these compounds maximizes
during daylight hours and minimizes during the night as would be expected.