# Peer review of "Molecular composition and volatility of isoprene photochemical oxidation secondary organic aerosol under low and high NOx conditions"

_Atmospheric Chemistry and Physics, 2016_

## Referee Comment (RC1) · Anonymous Referee #1 · 14 Sep 2016

General Comments

In general, this is a very well written manuscript focused on experiments that investigated secondary organic aerosol formation from isoprene oxidation under low humidity scenarios in which nitrogen oxide levels were varied. It includes an assessment of the molecular formulas of the constituents, as well as their volatility. A comparison is made to volatility (vapor pressure) prediction methodologies, as well as to recent field results. It is certainly a topic of much current interest in the atmospheric chemistry community and is appropriate for ACP. Scientifically, there are no major faults with the paper, as it is standard methodologies (chamber at PNNL, for example – please do not misinterpret, as I recognize the difficulty of doing chamber measurements) along side those

that are considered cutting edge (FIGAERO, for example) but evaluated and tested.

However, at times, I felt as if I were reading two papers that had been combined into one – that is, there could have been one manuscript that focused on these experiments with a comparison to field results; there are certainly sufficient data to do so. A second manuscript could have focused on the comparison of the volatility measurements to vapor pressure predictive capabilities. While the inability to predict vapor pressures of SOA constituents is not a new finding, the depth to which this area can be probed with the FIGAERO measurements is new and novel – perhaps warranting a second manuscript? This would certainly improve the readability of this manuscript. There were not significant issues with the writing, but the length and density of the manuscript made it slightly difficult to read.

Overall, I have no trouble recommending publication (following the authors addressing relatively minor specific comments below) based on its timeliness, topic, and high quality science. However, I would encourage both the editor and the authors to consider whether the manuscript could be split, simply to improve readability and perhaps to increase impact.

Specific Comments

It might be helpful to include a short figure that shows the chemical structure of specific molecules that are included in the discussion (ISOPOOH, ISOP(OOH)2, IEPOX, etc.) for the individuals who might not think about this on a daily basis.

Typo, p. 6, line 144, no need for the word "the"

Typo, p.7, line 168, "were" should be "was" (suite is singular)

Typo, p. 8, line 192, programmed has two m's

In Figure 1, it might be beneficial to show subsets that allow for seeing shorter periods of time. The length of time included/compressed on the x-axis makes seeing detail nearly impossible.

On page 10, lines 227 and 231. What is the basis for the factor of 1.5 to correct for wall losses?

Typo, p. 16, line 365, data suggest, not suggests

P. 19, third sentence of paragraph beginning on line 435 could be re-written for clarity.

On page 20, with the discussion of use of the measured Tmax to predict the Fp for comparison to measured Fp: Perhaps I am missing something, but this seems like a circular argument. It should be the case that a measured parameter that is related to Fp should do a better job predicting Fp than something unrelated like the vapor pressure estimation technique. I don't think that this adds very much to the paper. I do find the information that allows an investigation of predicted Fp as a function of molecular formula to be quite interesting and useful though.

On page 21, lines 473-475. Please reword for clarity. Which models predicted higher C*? And aren't group-contribution methods models? Perhaps word choice needs to be adjusted (or explain what the difference is).

Page 23, line 525. Does the change in vicosity imply a change in chemical composition? Do the measurements reflect that somehow?

On Figure 5, for predicted Fp (based on equation 1 and an estimated C* from group contribution methods), a COA is needed. What value was used here?

---

## Referee Comment (RC2) · Anonymous Referee #2 · 2 Oct 2016

General Comments

In this manuscript the authors report results of a study of the formation of aerosol from the OH oxidation of isoprene. Experiments were conducted in an environmental chamber under high and low NOx conditions, and gas and aerosol composition and particle volatility were analyzed using a FIGAERO-CIMS instrument with iodide ionization. A major focus was an evaluation of the effects of NOx on organic nitrate and SOA formation, gas-particle partitioning, and particle volatility. Information was also obtained on the possible role of oligomer formation on SOA composition and yield. The lab results were used to help interpret measurements made with the same instrument at the 2013 SOAS campaign, providing additional insight into the role of isoprene in aerosol

chemistry. The study was well done, including measurements, data analysis, and interpretation. The manuscript is also well written. Overall the manuscript is of high quality and represents an important contribution to atmospheric chemistry. I recommend it be published after the following comments are addressed.

Specific Comments

1. Line 160–162: Were the seed particles dried before entering the chamber. If not, might they not stay as deliquesced particles since the 50% RH is above the efflorescence point?

2. Lines 212–215: If wall loss of NO is significant would you expect wall loss of organic vapors to also be significant?

3. Lines 226–227: It would be useful to explain how you get the factor of 1.5 used to correct AMS data for particle wall loss.

4. I am aware of the mechanism for forming IEPOX under low NOx conditions, in which OH is lost from an –OOH group as the epoxide is formed. Is there an analogous mechanism for forming IEPOX under high NOx conditions in which NO2 is lost from a Âň–ONO2 group? If so it might be worth mentioning.

5. Lines 336–338: This sentence is confusing to me. Are you saying that HO2 reacts with an alkene C=C double bond? I don't think this happens.

6. Lines 344–348 and elswhere: What about the possibility that HNO3 serves as an acid catalyst under high NOx conditions?

7. Lines 394–395: I'm not sure what "included" means here. Do you mean the estimates have been corrected for these effects? Could make this clearer.

8. Lines 393–394. What are the estimated uncertainties in the measured values of Fp? 9. Lines 481–502: With regards to the possible presence of accretion reaction products: Do you observe any masses with identical profiles to $C_5H_{12}O_6$ or $C_5H_{10}O_6$

that would be indicative of co-products of the thermal decomposition reactions?

10. Lines 521–525: Is it likely that SOA can have high viscosity at the temperatures used for thermal desorption?

Technical Comments

1. Line 477: Should be "predicted".

2. Line 490: Should be FIGAERO-CIMS

3. Throughout the paper it seems like you switch arbitrarily from FIGAERO to FIGAERO-CIMS. I suggest you choose one and stick with it.

4. Line 618: Is the volatility of IEPOX really known to 5 significant figures?

---

## Author Comment (AC1) · 23 Nov 2016

*We thank the reviewer for their valuable comments. We have addressed each comment below in bold italic text, and indicate where necessary the corresponding changes we have made to the manuscript.*

Anonymous Referee #1

General Comments

In general, this is a very well written manuscript focused on experiments that investigated secondary organic aerosol formation from isoprene oxidation under low humidity scenarios in which nitrogen oxide levels were varied. It includes an assessment of the molecular formulas of the constituents, as well as their volatility. A comparison is made to volatility (vapor pressure) prediction methodologies, as well as to recent field results. It is certainly a topic of much current interest in the atmospheric chemistry community and is appropriate for ACP. Scientifically, there are no major faults with the paper, as it is standard methodologies (chamber at PNNL, for example – please do not misinterpret, as I recognize the difficulty of doing chamber measurements) along side those that are considered cutting edge (FIGAERO, for example) but evaluated and tested.

However, at times, I felt as if I were reading two papers that had been combined into one – that is, there could have been one manuscript that focused on these experiments with a comparison to field results; there are certainly sufficient data to do so. A second manuscript could have focused on the comparison of the volatility measurements to vapor pressure predictive capabilities. While the inability to predict vapor pressures of SOA constituents is not a new finding, the depth to which this area can be probed with the FIGAERO measurements is new and novel – perhaps warranting a second manuscript? This would certainly improve the readability of this manuscript. There were not significant issues with the writing, but the length and density of the manuscript made it slightly difficult to read.

Overall, I have no trouble recommending publication (following the authors addressing relatively minor specific comments below) based on its timeliness, topic, and high quality science. However, I would encourage both the editor and the authors to consider whether the manuscript could be split, simply to improve readability and perhaps to increase impact.

***We appreciate the referee's concern, and we have removed section 3.5 and figure 8 regarding the SOAS field data.***

Specific Comments

It might be helpful to include a short figure that shows the chemical structure of specific molecules that are included in the discussion (ISOPOOH, ISOP(OOH)2, IEPOX, etc.) for the individuals who might not think about this on a daily basis.

*We added a simple schematic showing isoprene and its major first and second generation oxidation products at both low and high-$NO_x$. High $NO_x$ arrows are highlighted in blue.*

Typo, p. 6, line 144, no need for the word "the"

*Fixed*

Typo, p.7, line 168, "were" should be "was" (suite is singular)

*Fixed*

Typo, p. 8, line 192, programmed has two m's

*Fixed*

In Figure 1, it might be beneficial to show subsets that allow for seeing shorter periods of time. The length of time included/compressed on the x-axis makes seeing detail nearly impossible.

*We added a supplemental online document with enlarged versions of figure 1 to show more detail. We prefer to show all the data collected in this case given the continuous flow approach used.*

On page 10, lines 227 and 231. What is the basis for the factor of 1.5 to correct for wall losses?

*The correction is determined by comparing the size distributions of ammonium sulfate particles in the inlet and exit flows of the chamber, prior to any SOA deposition. The size dependent particle wall-loss rate is then calculated based on the known chamber residence time and the difference in the number size distributions. The methods to determine the particle wall loss corrections are explained in more detail elsewhere [Liu et al., 2016] for this set of experiments. We refer readers to that paper more clearly now when discussing particle wall loss corrections.*

Typo, p. 16, line 365, data suggest, not suggests

*Fixed*

P. 19, third sentence of paragraph beginning on line 435 could be re-written for clarity.

*Reworded to:*

*However, the disagreement for $C_5H_{12}O_6$ (Fig. 5, bottom left, circles) cannot be explained solely by thermal decomposition of lower volatility material for two reasons: the thermogram shape is nearly Gaussian and therefore behaves like a single component, and the predicted $F_p$ is much larger than measured (opposite to the above situation).*

On page 20, with the discussion of use of the measured Tmax to predict the Fp for comparison to measured Fp: Perhaps I am missing something, but this seems like a circular argument. It should be the case that a measured parameter that is related to Fp should do a better job predicting Fp than something unrelated like the vapor pressure estimation technique. I don't think that this adds very much to the paper. I do find the information that allows an investigation of predicted Fp as a function of molecular formula to be quite interesting and useful though.

**We have tried to improve this section as the reviewer is perhaps missing what we think is an important distinction. The measured $F_p$ depends upon measured gas and particle phase concentrations of a compound the predicted $F_p$ depends upon (among other parameters) the C\* and total OA. In comparing measured and modeled $F_p$, there is an initial assumption that gas-particle equilibrium is achieved, and that the C\* is known. With the FIGAERO, we can independently constrain the C\* from the thermogram to use in the predicted $F_p$. That is, the temperature where a compound's signal reaches a maximum ($T_{max}$), is indicative of its C\* [Lopez-Hilfiker et al., 2014] but is largely independent of that compound's condensed-phase concentration. Thus using the FIGAERO measurement of C\* can improve the prediction of $F_p$ so that issues related to the validity of gas-particle equilibration timescales or the role of thermal decomposition during measurement of particle composition can be the focus of assessing causes of disagreement between theory and measurement (see [Lopez-Hilfiker et al., 2014; Lopez-Hilfiker et al., 2016] ). The following sentences have been added in a short succeeding paragraph to clarify its importance:**

*There are a number of possible reasons why measured and predicted $F_p$ do not agree, with one being that our methods to estimate C\* are flawed. This potential source of error can be addressed directly with the FIGAERO-CIMS thermogram, independently of the measured $F_p$, thereby allowing for a more robust assessment of whether (i) the assumption of gas-particle equilibrium is reasonable, (ii) there are possible biases in the measured $F_p$ due to thermal decomposition, or (iii) the C\* estimation methods are valid. The above analysis demonstrates that all possibilities arise in this data set.*

On page 21, lines 473-475. Please reword for clarity. Which models predicted higher C\*? And aren't group-contribution methods models? Perhaps word choice needs to be adjusted (or explain what the difference is).

*Agreed, group-contribution methods are models, although we use "method" here to refer specifically to the group-contribution calculations, while "model" refers to a continuum solvent model [Kurten et al., 2016] and 0-D box model [Valorso et al., 2011] which are more computationally expensive than the group-contribution methods. We have changed the wording throughout the text to be consistent, as well as altering this sentence to include the specific models:*

*Our observations suggest this assumption is incorrect, at least to the extent employed in these methods, and is supported by previous work that found group-contribution methods predicted significantly lower C\* than two computational models, the conductor-like screening model for real solvents (COSMO-RS) [Kurten et al., 2016] and the generator for explicit chemistry and kinetics of organics in the atmosphere (GECKO-A) [Valorso et al., 2011], did for compounds with multiple functionalities.*

Page 23, line 525. Does the change in viscosity imply a change in chemical composition? Do the measurements reflect that somehow?

*We note that the $T_{max}$ shift occurs for compounds with the same chemical composition and thus we assume that the compounds have the same structure. It is unclear why this $T_{max}$ shift occurs, and we can therefore only suggest possible explanations. We do not have direct measurements of the viscosity of the particles.*

On Figure 5, for predicted Fp (based on equation 1 and an estimated C\* from group contribution methods), a COA is needed. What value was used here?

*We reworded the sentence starting on line 418 to specify where $C_{OA}$ came from:*

*$F_p$ were predicted using equation 1 with $C_{OA}$ measured by the AMS and C\* calculated via the EVAPORATION group-contribution method [Compernolle et al., 2011], which generally gave similar estimates as the Capouet and Muller approach [2006].*

*Note: there was an error in the pie chart (figure 4) where we were previously not accounting for collection volume for the "other" portion of the pie. The updated figure does not change our conclusions.*

**References**

Capouet, M., and J. F. Muller (2006), A group contribution method for estimating the vapour pressures of alpha-pinene oxidation products, Atmospheric Chemistry and Physics, 6, 1455-1467.

Compernolle, S., K. Ceulemans, and J. F. Muller (2011), EVAPORATION: a new vapour pressure estimation methodfor organic molecules including non-additivity and intramolecular interactions, Atmospheric Chemistry and Physics, 11(18), 9431-9450, doi: 10.5194/acp-11-9431-2011.

Kurten, T., K. Tiusanen, P. Roldin, M. Rissanen, J. N. Luy, M. Boy, M. Ehn, and N. Donahue (2016), alpha-Pinene Autoxidation Products May Not Have Extremely Low Saturation Vapor Pressures Despite High O:C Ratios, J. Phys. Chem. A, 120(16), 2569-2582, doi: 10.1021/acs.jpca.6b02196.

Liu, J. M., E. L. D'Ambro, B. H. Lee, F. D. Lopez-Hilfiker, R. A. Zaveri, J. C. Rivera-Rios, F. N. Keutsch, S. Iyer, T. Kurten, Z. F. Zhang, A. Gold, J. D. Surratt, J. E. Shilling, and J. A. Thornton (2016), Efficient Isoprene Secondary Organic Aerosol Formation from a Non-IEPOX Pathway, Environmental science & technology, 50(18), 9872-9880, doi: 10.1021/acs.est.6b01872.

Lopez-Hilfiker, F. D., C. Mohr, M. Ehn, F. Rubach, E. Kleist, J. Wildt, T. F. Mentel, A. Lutz, M. Hallquist, D. Worsnop, and J. A. Thornton (2014), A novel method for online analysis of gas and particle composition: description and evaluation of a Filter Inlet for Gases and AEROsols (FIGAERO), Atmospheric Measurement Techniques, 7(4), 983-1001, doi: 10.5194/amt-7-983-2014.

Lopez-Hilfiker, F. D., C. Mohr, E. L. D'Ambro, A. Lutz, T. P. Riedel, C. J. Gaston, S. Iyer, Z. Zhang, A. Gold, J. D. Surratt, B. H. Lee, T. Kurten, W. W. Hu, J. Jimenez, M. Hallquist, and J. A. Thornton (2016), Molecular Composition and Volatility of Organic Aerosol in the Southeastern US: Implications for IEPOX Derived SOA, Environmental science & technology, 50(5), 2200-2209, doi: 10.1021/acs.est.5b04769.

Valorso, R., B. Aumont, M. Camredon, T. Raventos-Duran, C. Mouchel-Vallon, N. L. Ng, J. H. Seinfeld, J. Lee-Taylor, and S. Madronich (2011), Explicit modelling of SOA formation from alpha-pinene photooxidation: sensitivity to vapour pressure estimation, Atmospheric Chemistry and Physics, 11(14), 6895-6910, doi: 10.5194/acp-11-6895-2011.

---

## Author Comment (AC2) · 23 Nov 2016

*We thank the reviewer for their valuable comments. We have addressed each comment below in bold italic text, and indicate where necessary the corresponding changes we have made to the manuscript.*

Anonymous Referee #2

General Comments

In this manuscript the authors report results of a study of the formation of aerosol from the OH oxidation of isoprene. Experiments were conducted in an environmental chamber under high and low NOx conditions, and gas and aerosol composition and particle volatility were analyzed using a FIGAERO-CIMS instrument with iodide ionization. A major focus was an evaluation of the effects of NOx on organic nitrate and SOA formation, gas-particle partitioning, and particle volatility. Information was also obtained on the possible role of oligomer formation on SOA composition and yield. The lab results were used to help interpret measurements made with the same instrument at the 2013 SOAS campaign, providing additional insight into the role of isoprene in aerosol C1 chemistry. The study was well done, including measurements, data analysis, and interpretation. The manuscript is also well written. Overall the manuscript is of high quality and represents an important contribution to atmospheric chemistry. I recommend it be published after the following comments are addressed.

Specific Comments

1. Line 160–162: Were the seed particles dried before entering the chamber. If not, might they not stay as deliquesced particles since the 50% RH is above the efflorescence point?

*Yes, they passed through a diffusion drier before entering the chamber. The relative humidity of the drier output is routinely verified to be <30%, below the efflorescence point of ammonium sulfate particles [Cziczo and Abbatt, 2000]. We also verified that the seed particles were effloresced by comparing the size distribution of the particles exiting the 50% RH chamber.*

2. Lines 212–215: If wall loss of NO is significant would you expect wall loss of organic vapors to also be significant?

*This sentence was moved earlier in the paper into the "experimental overview" section from the "Gas-particle Partitioning: Saturation Vapor Concentrations and Oligomeric Content" section into the paragraph where particle wall loss is discussed:*

*Operating the chamber in continuous flow mode possibly reduces the net flux of organic compounds to the walls, at least for low volatility to semi volatile compounds, as some degree of equilibration can occur [Liu et al., 2016; Shilling et al., 2008].*

*Nonetheless, our vapor concentration data may be biased low due to some amount of loss to the walls.*

**Furthermore, we do not have evidence that a significant fraction of NO is lost to the walls, though it was not measured and Teflon is known to be somewhat permeable. We removed the last part of the sentence in question to prevent confusion that wall deposition is a main fate. NO is likely reacting fast enough with peroxy radicals that wall loss is not a significant fate.**

3. Lines 226–227: It would be useful to explain how you get the factor of 1.5 used to correct AMS data for particle wall loss.

**See our response to reviewer #1's comment above.**

4. I am aware of the mechanism for forming IEPOX under low NOx conditions, in which OH is lost from an –OOH group as the epoxide is formed. Is there an analogous mechanism for forming IEPOX under high NOx conditions in which NO2 is lost from a Ăˇn–ONO2 group? If so it might be worth mentioning.

**Good point, IEPOX can be formed from the reaction of isoprene hydroxy nitrate with OH [Jacobs et al., 2014; Lee et al., 2014; St Clair et al., 2016], albeit at fairly low overall yields. See response to comment #6, we added a sentence mentioning this possibility.**

5. Lines 336–338: This sentence is confusing to me. Are you saying that HO2 reacts with an alkene C=C double bond? I don't think this happens.

**Agreed. This sentence was changed to this:**

*Two possible sources of this compound are the reaction of the ISOPOOH derived peroxy radical ($C_5H_{11}O_6$) with $RO_2$, or a dihydroxy alkene undergoing reaction with OH and $O_2$ to form a dihydroxy peroxy radical, followed by reaction with $HO_2$.*

6. Lines 344–348 and elsewhere: What about the possibility that HNO3 serves as an acid catalyst under high NOx conditions?

**This is an interesting suggestion. Our two major tracers of IEPOX multiphase chemistry, $C_5H_{12}O_4$ and $C_5H_{10}O_3$, both decrease in concentration in the particle phase with increasing $NO_x$, arguing against $HNO_3$ acidifying the aerosol sufficiently to enhance "traditional" IEPOX SOA formation, but it may be driving some formation of oligomers from IEPOX. This idea would require more detailed experiments to assess. We have added the following sentences:**

*These two tracers are also observed in the $NO_x$ addition experiments performed in continuous-flow mode, though at lower concentrations. In these experiments there was not enough NO to completely suppress ISOPOOH formation and therefore IEPOX*

Responses to Referee #2                                                      2

*formation from ISOPOOH + OH. Moreover, IEPOX can also form, albeit at lower yields, from OH reactions with the first-generation isoprene hydroxy nitrate [Jacobs et al., 2014; Lee et al., 2014; St Clair et al., 2016]. However, the formation of IEPOX SOA tracers is more puzzling given the very low reactive uptake of IEPOX expected on solid inorganic seed coated with isoprene SOA [Gaston et al., 2014; Riedel et al., 2015]. Perhaps nitric acid catalyzed IEPOX multiphase chemistry is contributing to the formation of these tracers at high $NO_x$. That said, observation of these tracers, also in the absence of $NO_x$ addition, indicates that this explanation not sufficient.*

7. Lines 394–395: I'm not sure what "included" means here. Do you mean the estimates have been corrected for these effects? Could make this clearer.

***See 8. below. This paragraph was slightly rearranged and sentences reworded to make the error corrections more clear.***

8. Lines 393–394. What are the estimated uncertainties in the measured values of Fp?

***The measured $F_p$ values contain errors primarily from losses of gaseous vapors to the chamber walls as well as losses in the sampling inlet. Because the sampling inlet was short (1 m) and the flow volume was high (12 lpm) for the data presented in this plot, we expect inlet losses to be at most 25-44% assuming a diffusivity of 0.05-0.1 $cm^2$ $s^{-1}$ and irreversible loss at the wall. We made this more explicit in the text:***

*The largest source of error in the measured $F_p$, beyond thermal decomposition during desorption, is diffusion-controlled vapor losses in the inlet which were corrected for by assuming a diffusivity of 0.05-0.1 $cm^2$ $s^{-1}$, although the variability is not discernable on the log scale. Inlet losses are 25-44% for the range of diffusivities, a small effect on the comparison of measured and predicted $F_p$ as we show below.*

9. Lines 481–502: With regards to the possible presence of accretion reaction products: Do you observe any masses with identical profiles to C5H12O6 or C5H10O6 that would be indicative of co-products of the thermal decomposition reactions?

***On lines 445-460 we discuss the reasons these specific compounds do not show evidence of being thermal decomposition products. We added "as opposed to thermal decomposition" to the end of the sentence on line 459-460to make it clearer. We also added a paragraph to the end of section 3.3 and a corresponding figure (now Fig. 6) indicating evidence of accretion products measured by the FIGAERO-CIMS, as well as adding the sum of signal from compounds having 6 or more carbons to the bottom left panel of Figure 7 (previously Fig. 6).***

10. Lines 521–525: Is it likely that SOA can have high viscosity at the temperatures used for thermal desorption?

*How isoprene SOA viscosity varies as a function of temperature remains poorly studied, though our expectation is that viscosity lowers significantly as the aerosol is heated to 200 $^0$C. It would be interesting to examine the thermal desorption behavior of SOA as a function of known viscosity but this has not been done. It is possible that there are two phases present, one with low viscosity that persists to temperature higher than 100 $^0$C. However, we have no evidence for two organic phases.*

Technical Comments

1. Line 477: Should be "predicted".

*Fixed*

2. Line 490: Should be FIGAERO-CIMS

*Fixed*

3. Throughout the paper it seems like you switch arbitrarily from FIGAERO to FIGAERO-CIMS. I suggest you choose one and stick with it.

*We changed all "FIGAERO" references to "FIGAERO-CIMS" except where specifically talking about the unit itself.*

4. Line 618: Is the volatility of IEPOX really known to 5 significant figures?

*Good catch. We have reduced the number of significant figures to 1, more in line with previously stated c\*'s throughout the text, and reduced all other group-contribution method obtained c\*'s to 1 significant figure as well.*

*Note: there was an error in the pie chart (figure 4) where we were previously not accounting for collection volume for the "other" portion of the pie. The updated figure does not change our conclusions.*

**References**

Cziczo, D. J., and J. P. D. Abbatt (2000), *Infrared observations of the response of NaCl, MgCl2, NH4HSO4, and NH4NO3 aerosols to changes in relative humidity from 298 to 238 K, J. Phys. Chem. A, 104(10), 2038-2047, doi: 10.1021/jp9931408.*

Gaston, C. J., T. P. Riedel, Z. F. Zhang, A. Gold, J. D. Surratt, and J. A. Thornton (2014), *Reactive Uptake of an Isoprene-Derived Epoxydiol to Submicron Aerosol Particles, Environmental science & technology, 48(19), 11178-11186, doi: 10.1021/es5034266.*

Jacobs, M. I., W. J. Burke, and M. J. Elrod (2014), *Kinetics of the reactions of isoprene-derived hydroxynitrates: gas phase epoxide formation and solution phase hydrolysis, Atmospheric Chemistry and Physics, 14(17), 8933-8946, doi: 10.5194/acp-14-8933-2014.*

Lee, L., A. P. Teng, P. O. Wennberg, J. D. Crounse, and R. C. Cohen (2014), *On rates and mechanisms of OH and O3 reactions with isoprene-derived hydroxy nitrates, The journal of physical chemistry. A, 118(9), 1622-1637, doi: 10.1021/jp4107603.*

Liu, J. M., E. L. D'Ambro, B. H. Lee, F. D. Lopez-Hilfiker, R. A. Zaveri, J. C. Rivera-Rios, F. N. Keutsch, S. Iyer, T. Kurten, Z. F. Zhang, A. Gold, J. D. Surratt, J. E. Shilling, and J. A. Thornton (2016), *Efficient Isoprene Secondary Organic Aerosol Formation from a Non-IEPOX Pathway, Environmental science & technology, 50(18), 9872-9880, doi: 10.1021/acs.est.6b01872.*

Riedel, T. P., Y. H. Lin, H. Budisulistiorini, C. J. Gaston, J. A. Thornton, Z. F. Zhang, W. Vizuete, A. Gold, and J. D. Surrattt (2015), *Heterogeneous Reactions of Isoprene-Derived Epoxides: Reaction Probabilities and Molar Secondary Organic Aerosol Yield Estimates, Environ. Sci. Technol. Lett., 2(2), 38-42, doi: 10.1021/ez500406f.*

Shilling, J. E., Q. Chen, S. M. King, T. Rosenoern, J. H. Kroll, D. R. Worsnop, K. A. McKinney, and S. T. Martin (2008), *Particle mass yield in secondary organic aerosol formed by the dark ozonolysis of alpha-pinene, Atmospheric Chemistry and Physics, 8(7), 2073-2088, doi: 10.5194/acp-8-2073-2008.*

St Clair, J. M., J. C. Rivera-Rios, J. D. Crounse, H. C. Knap, K. H. Bates, A. P. Teng, S. Jorgensen, H. G. Kjaergaard, F. N. Keutsch, and P. O. Wennberg (2016), *Kinetics and Products of the Reaction of the First-Generation Isoprene Hydroxy Hydroperoxide (ISOPOOH) with OH, J. Phys. Chem. A, 120(9), 1441-1451, doi: 10.1021/acs.jpca.5b06532.*